# Quantifying coastal freshwater extremes during unprecedented rainfall using long timeseries multi-platform salinity observations

Neil Malan [1,2] ✉, Moninya Roughan [1,2], Michael Hemming[1,2] & Tim Ingleton [3]

During 2022, extreme rainfall occurred across southeast Australia, making it the wettest year on record. The oceanic impact of extreme rainfall events in normally 'dry' regions is not well understood, as their effects are challenging to observe. Here, we use unique multi-platform timeseries and spatial data from 36 autonomous ocean glider missions over 13 years, and we define an extreme salinity threshold inshore of the East Australian Current. We show that the freshwater plume extended fivefold further than previously thought. The compound effect of multiple large rainfall events resulted in a newly observed stratification ('double-stacking') dynamic, with the stratification being largely controlled by salinity. Extreme salinity events are known to be important for species composition of local fisheries as well as detrimental for coastal water quality. Such events and their impacts may become more common as extreme rainfall events are projected to become more frequent in a changing climate. Hence, comprehensive observing strategies facilitating identification of salinity extremes are essential.

Unprecedented La Niña conditions began in September 2020 and continued through to late 2022, resulting in the first 'triple-dip' La Niña in the 21st century[1]. Over southeastern Australia, La Niña conditions are typically associated with above average rainfall conditions[2,3] due to increased supply of evaporative moisture available for precipitation[3]. In 2022, extreme rainfall resulted in severe, unprecedented, widespread and repeated flooding throughout coastal areas of southeastern Australia, with damage estimated at over a billion Australian dollars. The effect of La Niña on rainfall in southeastern Australia is expected to intensify under future conditions, in line with the expected global increase in extreme events associated with climate change[4,5] meaning that extreme wet years may become more common.

While there is a vast literature on the effect of extreme rainfall events on land, less is known about the impact of extreme rainfall in the coastal ocean. This is despite 80% of global surface freshwater fluxes taking place over the ocean[6]. Our lack of knowledge of freshwater's impact on coastal circulation is even more marked outside the well known large river plumes of the world such as the Amazon[7], Mississippi[8], Changjiang[9] or Columbia[10]. Exploration of the subsurface extent and 3-dimensional structure of low-salinity plumes associated with continental freshwater runoff is limited by a global lack of in-situ observations in shelf seas (waters shallower than 1000 m). The observational problem is exacerbated inshore of the powerful poleward-flowing western boundary currents such as the East Australian Current (EAC), where strong advection from offshore currents increases the variability in flow regimes on the continental shelf, making it challenging to observe.

Compared to other regions, rainfall in southeastern Australia is generally low and the large estuaries such as the Hawkesbury River are tidally-influenced drowned river valleys[11] with low freshwater

[1]Coastal and Regional Oceanography Lab, School of Biological Earth and Environmental Sciences, UNSW Sydney, Sydney 2052 NSW, Australia. [2]Centre for Marine Science and Innovation, UNSW Sydney, Sydney 2052 NSW, Australia. [3]Waters, Wetlands and Coastal Science, New South Wales Department of Planning and Environment (DPE), Sydney 2000 NSW, Australia. ✉e-mail: n.malan@unsw.edu.au

discharge. Knowledge of river plume spreading and freshwater dynamics in shelf waters is limited to a single idealised modelling study[12] and freshwater impacts have typically been assumed negligible in studies of the ocean dynamics of the region[13–15].

With the recent emphasis on marine extremes, particularly the growing interest in marine heatwaves[16–20], a framework has been established for the development of climatologies that facilitate the ready identification of extreme events using percentile outliers. This framework is now frequently applied to temperature, but has not been broadly used to identify salinity extremes and thus enable the quantification of their impacts, due to lack of appropriate data in most regions.

Since 2008, a comprehensive ocean observing system has been under development around Australia through their Integrated Marine Observing System (IMOS). Consequently, the productive Hawkesbury shelf region (32.5–34°S)[21], inshore of the East Australian Current, is one of the best observed western boundary current shelves globally. In this region routine monitoring by in-situ moorings and autonomous underwater gliders provides a full-depth picture of temperature, salinity and optical properties at a high spatial and temporal resolution over the continental shelf[22,23]. Gliders have the advantage that they can sample autonomously even during extreme ocean weather events[24] or under ice[25], and provide 3-D spatial coverage of the coastal ocean. When combined with moored timeseries data they provide comprehensive data coverage in both time and space.

With over a decade of sustained observations of salinity off southeastern Australia we can develop a daily climatology and therefore systematically identify and characterise extreme low salinity (or freshwater) events in coastal waters. Studies such as these are important for understanding the physical and ecological impacts of marine extremes in a changing climate.

In this study, we use a multi-platform approach, leveraging satellite data, long timeseries from an coastal oceanographic mooring, estuarine loggers, and data from 36 ocean glider missions obtained over a decade to identify and quantify the impact of extreme freshwater discharge events over the continental shelf. The study region lies inshore of an energetic western boundary current, the East Australian Current, and offshore from the most densely populated areas of the southeastern Australia coast. We observe the impact of sustained extreme rainfall on the coastal ocean during 2022, the wettest year on record, and we contrast it with a large rainfall event during 2015, a year of average rainfall. We also contrast the impact of large rainfall events on the coastal ocean with drier conditions that usually prevail in this region. Our approach is readily applicable to other well observed regions globally and provides a framework for identification of extreme freshwater events in shelf waters.

## Results
### Defining extreme salinity events
We define extreme low salinity events as those below the daily varying 5th percentile at the PH100 coastal mooring (Fig. 2c) using an approach analogous to the framework used to define marine heatwaves[26]. The 5th percentile was chosen as the best threshold for extreme events as it separates the long 'tail' of extreme events from the expected variability, when considering the distribution of salinity in the dataset (Fig. 2c). The use of a seasonally varying percentile threshold, rather than a fixed one, as well as the use of the observed salinity distribution to guide the choice of that percentile threshold, is important in choosing appropriate extreme salinity values for the region. In a region of low salinity variability, a relatively small change in salinity could be considered extreme, while in a region of high salinity variability a larger drop in salinity would be needed to be categorised as extreme. As suggested by[26], we ignore intermittent periods of 2 days or less when salinity is less anomalous and followed by another extreme low salinity event, considering the whole period to be part of the same event. For

more information on the salinity climatology and extreme categorisation, refer to the methods section.

### Relationship between rainfall, estuarine, and coastal salinity
In order to infer the impact of extreme rainfall on the coastal ocean, we first explore whether extreme rainfall events over land indeed coincide with low salinity in the coastal ocean, via the increased flow of freshwater out of estuaries and rivers. In the Hawkesbury shelf region, data from 2022 shows a significant correlation (r = 0.73 at a 1-day lag) between the salinity at the mouth of the Hawkesbury River (the largest river in the region) and salinity at the PH100 mooring site, approximately 65 km away. During 2022 periods of low estuarine salinity (Fig. 3a) coincide with low coastal salinity (Fig. 3b) and occur after large rainfall events, which take place in March, April, and July (Fig. 3c).

### Case studies
2022 was the wettest year on record for much of the southeast coast of Australia, with cumulative rainfall at Sydney Airport of more than 2 m being recorded (almost double the annual average, Fig. 2a), with intense and widespread flooding resulting in the large discharge of freshwater into the coastal ocean (Fig. 1b). The period from late February to early May featured particularly heavy rainfall with almost a metre of accumulated rainfall in that period alone. In contrast to 2022, there was below-average rainfall in 2018, after an El Niño period preceded large bushfires in eastern Australia[27]. The contrasting oceanic conditions between these two years, 2022 (large rainfall, flooding) and 2018 (below average rainfall, bushfires), can be seen in the salinity timeseries from the PH100 mooring (Fig. 2b). Near-surface salinity at PH100 during January to May 2018 was consistently above average, while salinity during the same period in 2022 was below average most of the time with an almost unbroken period of extreme low salinity from the beginning of March to the end of April.

A timeseries of rainfall (Fig. 2d) from Sydney Airport (location shown in Fig. 1a), shows the large rainfall events that occurred in the first half of 2020, 2021 and 2022. Each of these (as well as an earlier large rainfall event in April 2015) have been sampled by the regular IMOS glider missions over the Hawkesbury shelf. Hence, from all 36 available glider missions (see Fig. 2d for their temporal distribution), we categorise 4 missions as having captured data during 'wet' periods, defined by having consistent extreme (using the daily climatology at the PH100 mooring) low salinity in the top 50 m of the water column. Figure 1a shows the tracks and timing of these 4 missions in the context of all historical missions. The glider mission measuring the all-time lowest salinity took place in April 2022, immediately after a very large rainfall event (Fig. 2a,d, Fig. 3c).

Remarkably, this April 2022 glider sampled salinity below the daily-varying 5th percentile for April (35.16 psu) during every profile of its 400 km long transect along the Hawkesbury shelf (Fig. 4a), with extreme salinity (below 5th percentile) encountered offshore of the 200 m isobath, more than 70 km from the coast. The water column was even fresher further south, with a minimum salinity of 32.9 psu recorded at 34°S. An across-shelf section at 33.5°S (Fig. 4b) shows that the stratification of salinity in the water column is relatively consistent in the cross-shelf direction. An initial lower salinity (<34.7 psu) layer of water occupies the top 20 m of the water column. Below this, salinity increases, but remains below the 35.16 psu extreme salinity threshold until a depth of 50 m where the normal shelf water mass is present, marked by salinity above 35.4 psu. Temperature is well-mixed down to a depth of 50 m (Fig. 4c). The buoyancy frequency ($N^2$, Fig. 4d) reveals a mostly stable water column ($N^2 > 0$), with two distinct levels of strong buoyancy gradients, at 20 m depth associated with the first freshwater layer, and at 50 m depth, associated with both the temperature stratification and the halocline caused by the second freshwater layer.

We use coloured dissolved organic matter (CDOM) as measured by the glider as a marker for the location of river plumes. A spatial view of the distribution of CDOM (Fig. 4f) reveals that, similarly to salinity, CDOM increases as the glider travels polewards, but is patchier than other parameters with higher CDOM associated with proximity to estuaries. The cross-shelf section at 33.5°S (Fig. 4e) is located near the mouth of the Hawkesbury River, one of the largest estuaries in the region. The high (>1.2 mg m$^{-3}$) CDOM extends throughout the upper 20 m of the water column out to 20 km of the glider's track offshore (Fig. 4e), whereupon it subducts beneath warmer less dense offshore water.

We contrast the large rainfall event during the wettest year on record (April 2022) with a case study of a large rainfall event during an average rainfall year 2015 (Fig. 2a). In both cases rainfall in the two weeks preceding the glider mission is similar (255 mm in 2015 and 311 mm in 2022), but the rainfall in the 3 months preceding is vastly different (479 mm in 2015 and 1125 mm in 2022).

In April 2015, glider data showed extreme low salinity poleward of approximately 32.75°S (Fig. 5a), with only a single layer of low salinity water extending across the shelf (Fig. 5b), which has far more cross-shelf variability in stratification than the previous April 2022 case study. The extreme low salinity layer extends to a depth of 40 m for the first 10 km from the coast, but then shallows to 10 m approximately 50 km offshore. At the shelf break the low salinity layer is subducted, resulting in a subsurface 20 m thick layer of fresher water at 50 m depth below the warm (Fig. 5c) salty EAC water offshore. Unlike the 2022 case study, the April 2015 glider did not observe the double layering of density gradients in N$^2$ (Fig. 5d), whilst CDOM follows the same pattern of cross-shelf variability as seen in salinity (Fig. 5e, f).

The mesoscale context of these two wet event case studies is shown using surface maps of satellite geostrophic velocities and ocean colour (Figs. 4g and 5g). Both case studies show locally elevated ocean colour close to the coast in the so-called 'green ribbon'[28,29] and a strong (>1 m.s$^{-1}$) southwestward EAC jet offshore. Both case studies also show a strong front between the warm, salty EAC waters offshore, and the cooler shelf water, which has been freshened by the large rainfall events. This front is well represented in the ocean colour images of chlorophyll, dividing the shelf waters (high chlorophyll) from the offshore EAC water (low chlorophyll). In the case of April 2015 this front is associated with a streamer of fresher water (the glider crosses it some 60 km offshore, Fig. 5b,g) that is high in both CDOM and chlorophyll. Remarkably, this streamer extends polewards from 33 to 34.5°S, and eastwards from 152 to 152.7°E (Fig. 5g).

## Effect of freshwater outflows on coastal circulation

To quantify the impact of estuarine freshwater discharge on the coastal circulation, we calculate the buoyancy-driven flow caused by the freshwater plume observed by the glider during the April 2022 case study. Using the horizontal density gradients in the top 50 m of the water column observed from the glider section, from near the coast to the 200 m isobath, (Fig. 4g) and a logger deployed in the mouth of the Hawkesbury River, the buoyancy-driven flow can be estimated using the Heaps formulation of the momentum equations[30,31]. These theoretical buoyancy-driven flows are then compared to the surface and depth-integrated velocities measured directly by the glider (Fig. 6a). Inshore of the 100 m isobath, there is strong agreement in direction and to a lesser extent magnitude between the buoyancy-driven and observed velocities, which both flow northeastward at up to 0.5 m s$^{-1}$. At the 100 m isobath (15 km offshore), there is a surface velocity front

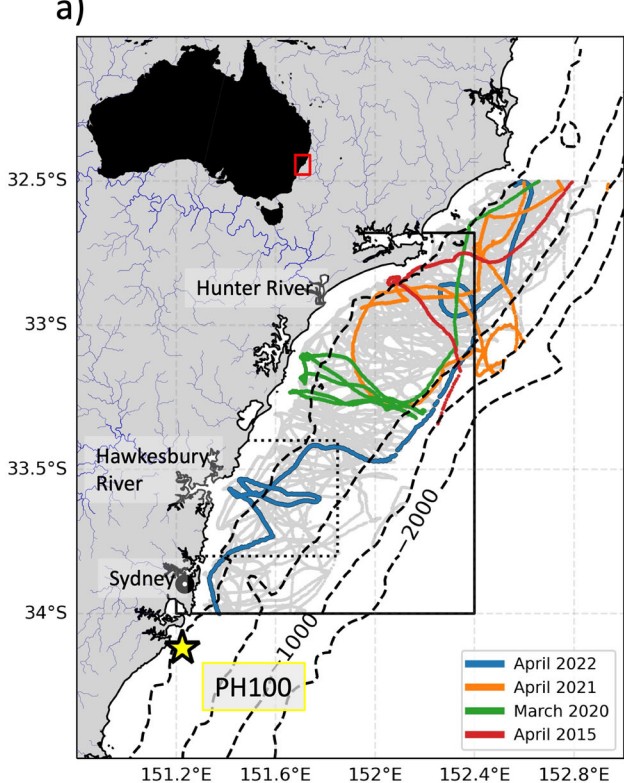

a)

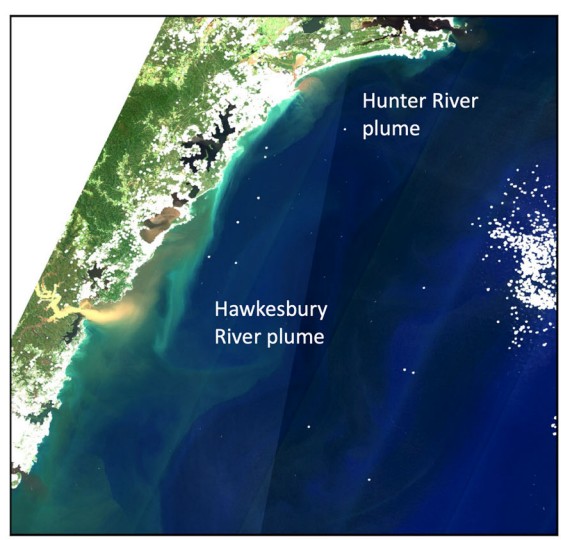

b)

**Fig. 1 | Summary of the study region and location of glider tracks. a** Map of Hawkesbury shelf region, showing glider tracks during both normal (grey) and extreme low salinity (colours) events, the yellow star shows the location of the PH100 mooring site, while major rivers (scaled by discharge volume) are shown in blue. The location of the Sydney Airport rainfall station is marked by the black circle. The black dotted box shows the domain of Fig. 6 and the solid black box shows the domain of panel **b**), a visible colour image taken from the Sentinel-2 satellite on 10 April 2022 showing the Hawkesbury and Hunter River plumes and discolouration of shelf waters associated with a large rainfall event.

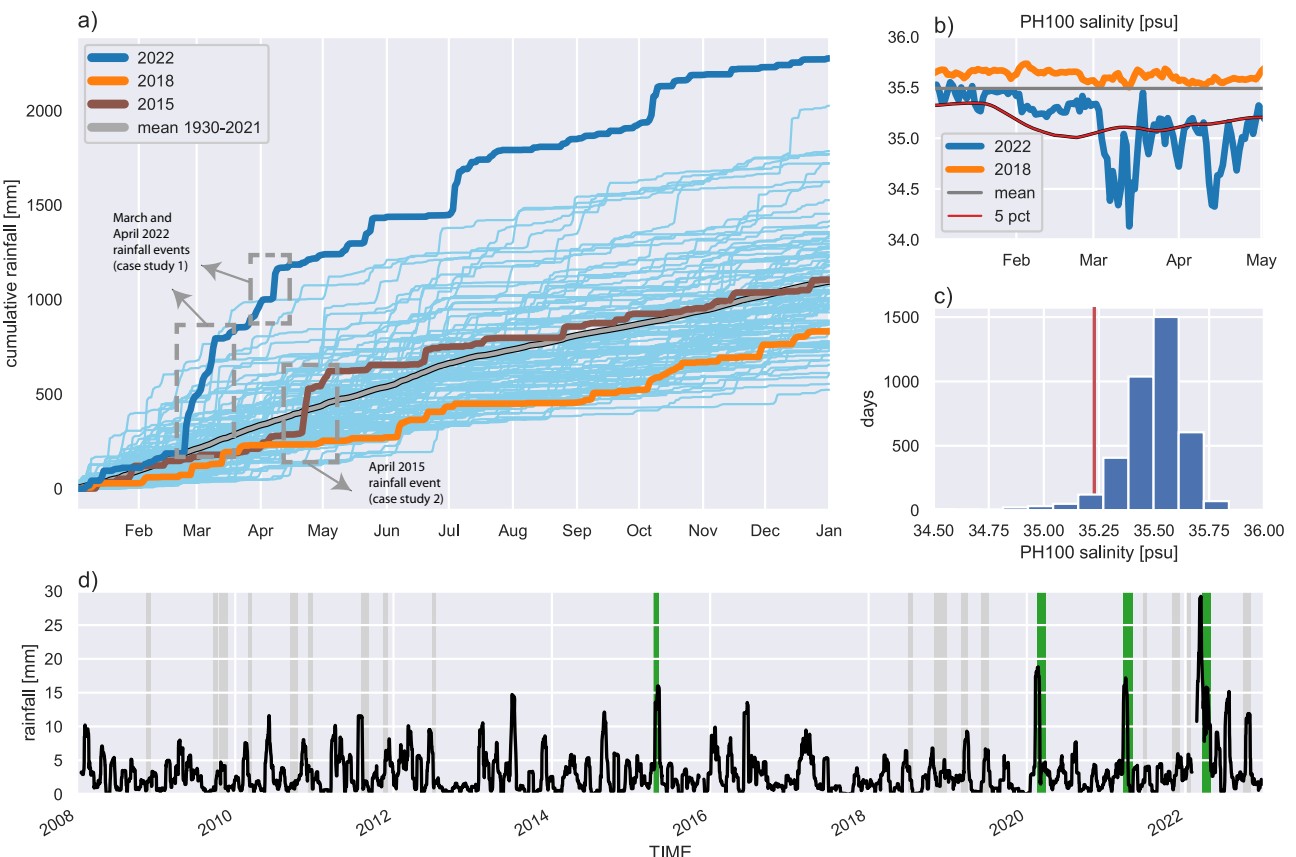

**Fig. 2 | Rainfall and salinity timeseries from Sydney airport and the PH100 mooring. a** Cumulative annual rainfall from Sydney Airport (see Fig. 1 for location), **b** Salinity timeseries at depths between 15 and 24 m for January to May at the PH100 mooring site (see Fig. 1 for location), showing the mean, daily 5th percentile and the years 2018 and 2022. **c** Distribution of daily salinity at PH100 between 2008 and 2022, vertical red line shows the annual 5th percentile. **d** 3-week running mean of rainfall at Sydney Airport from Jan 2008-Dec 2022. Green and grey bars show timing of the extreme low and normal salinity glider missions, respectively.

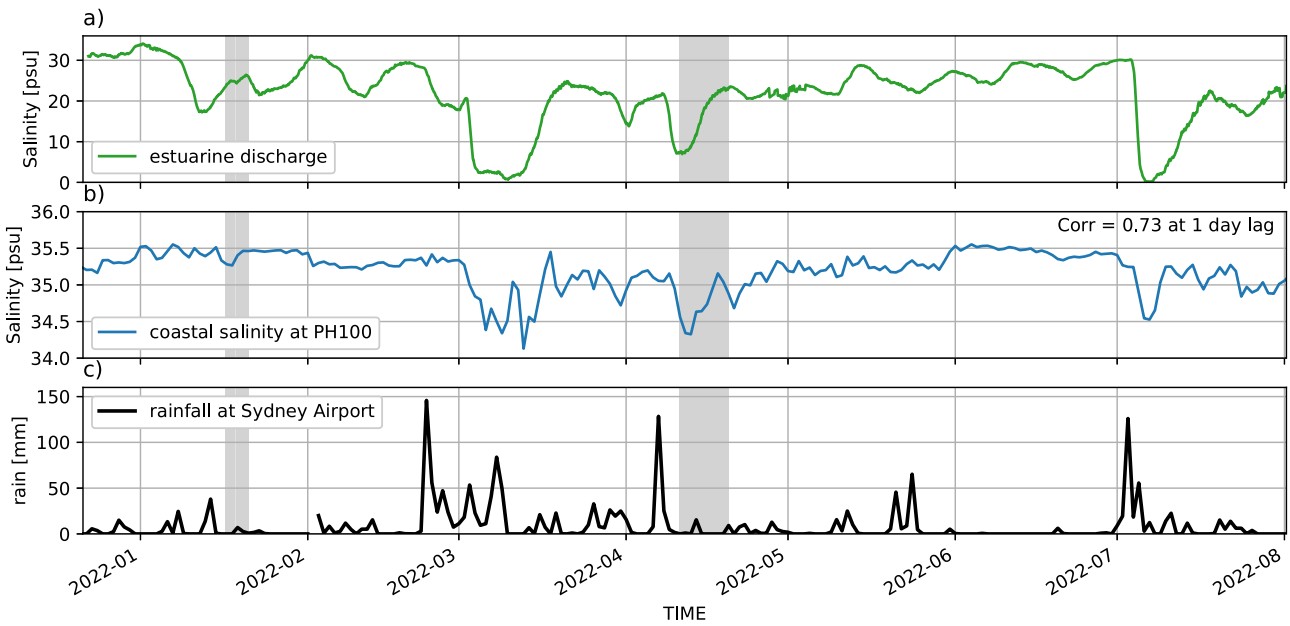

**Fig. 3 | A comparison on estuarine and offshore salinity timeseries from January to August 2022. a)** surface salinity in the mouth of the Hawkesbury River estuary, **b)** near-surface salinity at the PH100 mooring site and **c)** rainfall at Sydney Airport. Grey shading shows the time extent of glider missions in the Hawkesbury shelf region during 2022.

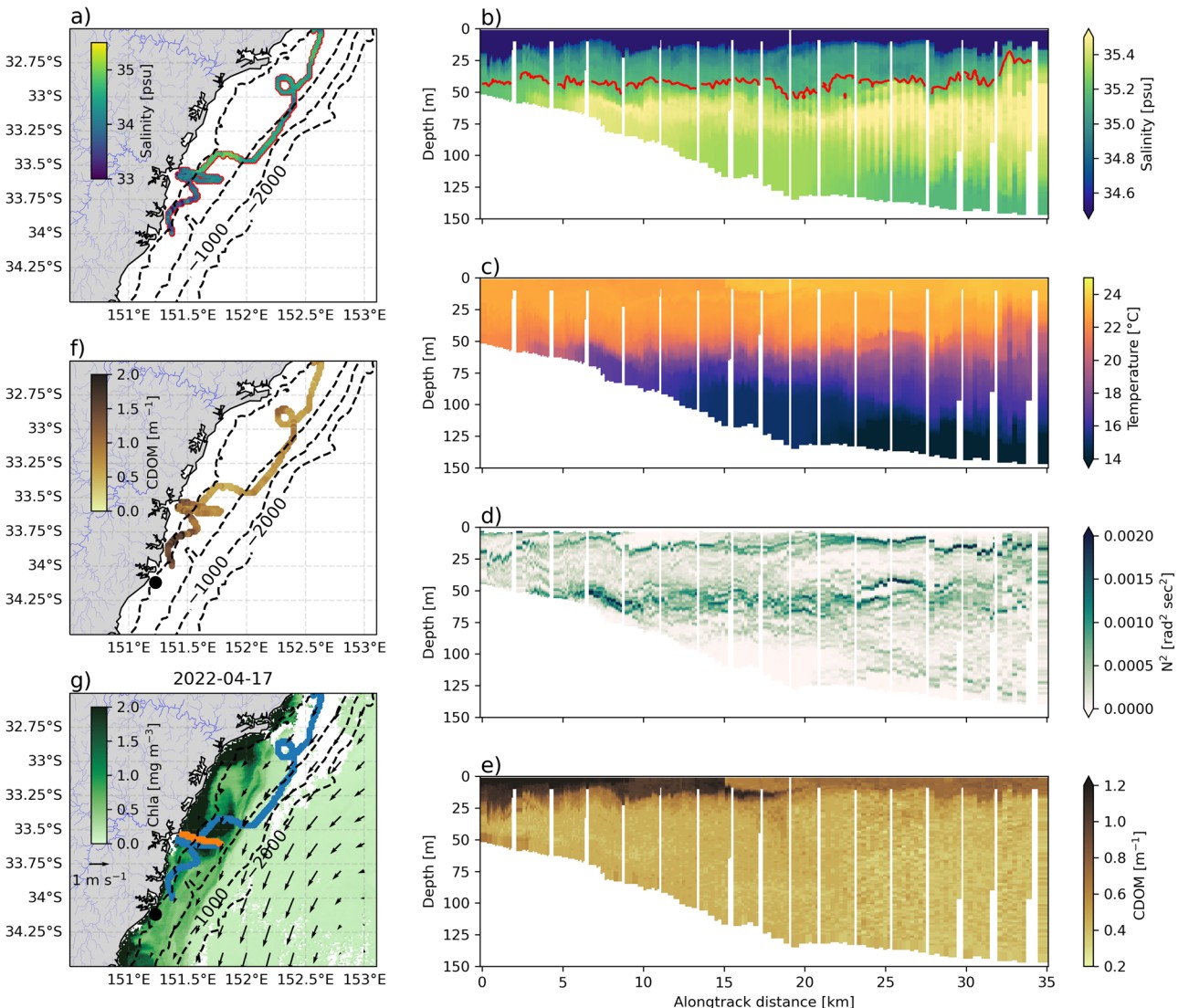

**Fig. 4 | Case study of April 2022 extreme wet event. a** Glider track with minimum salinity for each profile (colours), and red outline for profiles where extreme low salinity water is present. Cross-shelf sections at 33°S for **b**) Salinity, with the red contour showing the extreme low salinity threshold **c**)Temperature **d**) N² **e**) and coloured dissolved organic matter (CDOM). **f** Shows the glider track coloured by the maximum CDOM value for each profile. **g** Shows surface maps of daily satellite-derived ocean colour (shading) and geostrophic velocities (vectors), glider tracks are shown in blue, with the cross-shelf section shown in orange. Note that along-track distance is in the across-shelf direction.

which coincides with a sharp increase in temperature (Fig. 4c) and decrease in CDOM (Fig. 4e). This points toward this front marking the boundary between the freshwater plume (cool water flowing north-eastward) and water entrained by the EAC (warmer water flowing southwestward along the shelf edge). Full-depth geostrophic velocities (Fig. 6b), are also calculated from the glider density field (Fig. 6c) to show the vertical structure. Although the utility of geostrophy can be limited this close to the coast, we see that the northeastward inshore flow associated with the freshwater plume, while surface-intensified, extends to a near-bottom depth of 50 m. The northeastward plume flow shallows and weakens in the offshore direction. The effect of the poleward flowing EAC can be seen at the very offshore end of the transect (Fig. 6b).

### Spatial extent of low salinity impact

The offshore extent of low salinity conditions associated with the 2015 and 2022 case studies can also be observed using the monthly SMOS satellite sea surface salinity product (Fig. 7). In April 2022 (Fig. 7a), low salinity water (<35 psu) extends southwestward along the offshore edge of the shelf break from 33°S to 36°S. In contrast, in April 2015, a plume of lower salinity water extends in the opposite direction, northeastwards along the shelf edge from 33°S to 31°S (Fig. 7b). It should be noted that salinity offshore of the shelf break is generally fresher than the time-mean (Fig. 7c) in 2022 and saltier than the mean in 2015. This interannual variability is clear in a timeseries of satellite salinity extracted from a box along the shelf edge of the Hawkesbury shelf (Fig. 7d). We see that 2015 has the highest salinity in the 7-year record, while the salinity in 2022 is anomalously low (>0.6 psu below the time-mean), although the uncertainty associated with this dataset must be acknowledged (Fig. 7d). The difference in salinity between 2015 and 2022 is consistent with the amount of rainfall received in the first 3 months of each year, with accumulated rainfall from January to March being 4 times greater in 2022 compared to 2015 (Fig. 2a).

### Contribution of extreme low salinity water to vertical stratification

Our results show that the discharge of freshwater into the coastal ocean as a result of extreme rainfall events has impacted both the

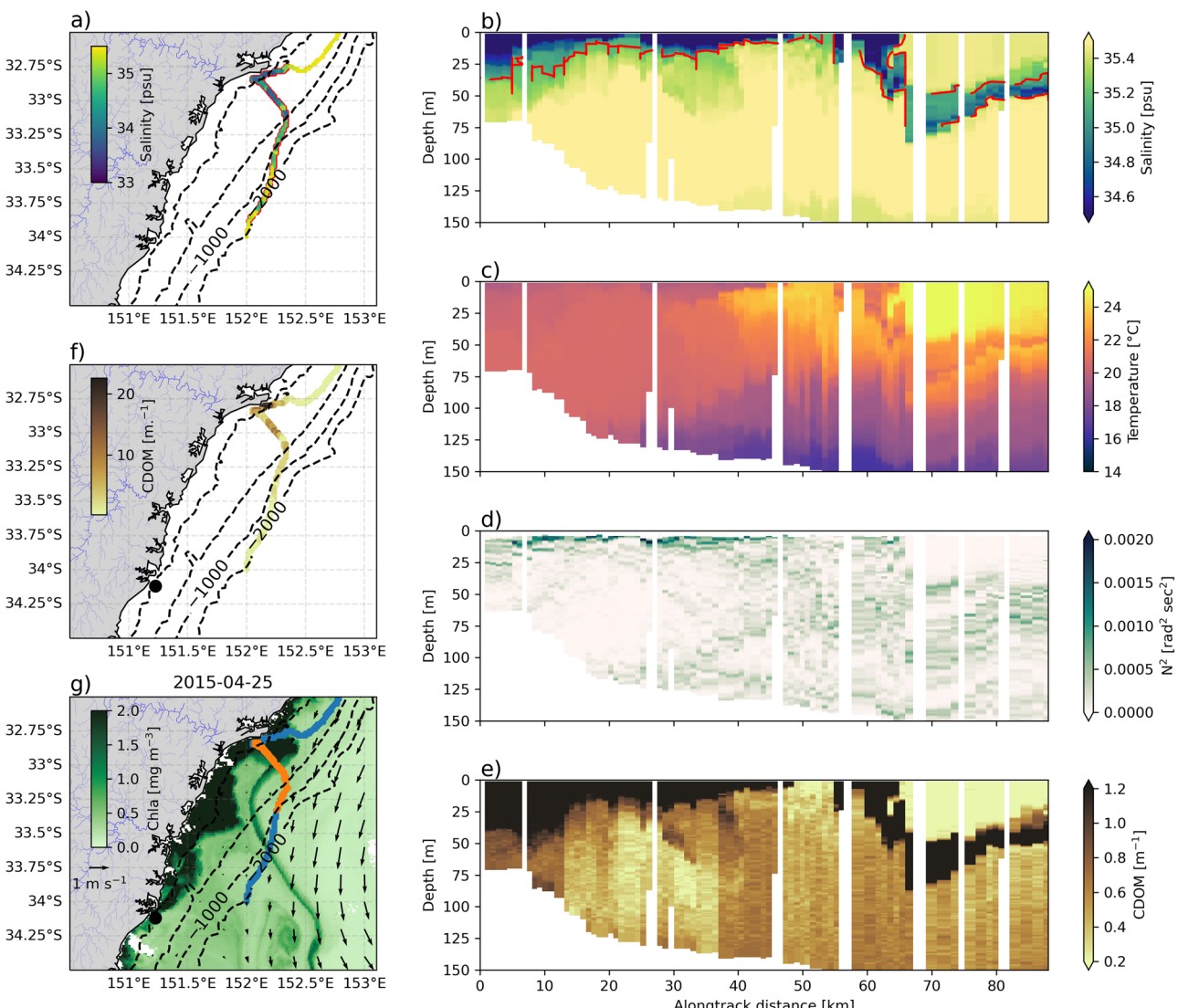

**Fig. 5 | Case study of April 2015 extreme wet event. a** Glider track with minimum salinity for each profile (colours), and red outline for profiles where extreme low salinity water is present. Cross-shelf sections at 33.5°S for b) Salinity, with the red contour showing the extreme low salinity threshold **c**)Temperature **d**) N² **e**) and coloured dissolved organic matter (CDOM). **f** Shows the glider track coloured by the maximum CDOM value for each profile. **g** Shows surface maps of daily satellite-derived ocean colour (shading) and geostrophic velocities (vectors), glider tracks are shown in blue, with the cross-shelf section shown in orange. Note that along-track distance is in the across-shelf direction.

water mass properties and circulation of the coastal ocean in the Hawkesbury shelf region. In addition, the dynamics of the vertical stratification of the water column has also been affected by the extreme low salinity events in 2022 and 2015. We quantify this effect using the Stratification Control Index[32], which compares the role of temperature and salinity in the overall stratification of the water column. Usually the Hawkesbury shelf (and greater EAC) region is assumed to be temperature stratified, which we confirm by a mean section of glider transects previously collected at 33.5°S (Fig. 8a). However, in the 2022 case study (13 April glider transect, Fig. 8b) the upper 30-50 m of the water column are dominantly stratified by salinity, with a temperature stratified layer below at 40-70 m. These layers extend across the width of the shelf. In the April 2015 case study (Fig. 8c), the boundary between temperature and salinity stratified water was not as stark and the vertical layering not as consistent when compared with 2022, with more salinity stratified water close to the coast, and temperature stratified water offshore. Although, in both cases the extreme low salinity waters resulting from heavy rainfall had altered the stratification of the shelf water mass.

**Comparison with dry years**

The April 2022 and April 2015 case studies presented above give insight into the way in which large rainfall events impact the Hawkesbury shelf during high and average rainfall years, respectively. However, it is also valuable to compare composite means of the 4 'wet' missions, which take place during large rainfall events to the other 36 glider missions, which take place during the 'dry' conditions of low freshwater discharge more common in southeast Australia (Fig. 9). As the effects of freshwater discharge on shelf waters appear to operate over long length scales (~30–100 km, Figs. 4a and 5a), we take the mean of each glider mission during the time that it is in the region of interest (32.5–34.5°S and inshore of the 200 m isobath).

The impact of the large rainfall event associated with the 'wet' missions is immediately apparent in the composite mean of salinity, with surface salinity during 'wet' missions being an average 0.75 psu fresher. The difference is highest at the surface, but is detectable in the composite mean down to a depth of 100 m (Fig. 9a). Temperatures during 'wet' missions are also warmer (Fig. 9b), leading to the water column being less dense (Fig. 9c), particularly in the upper 60 m. The

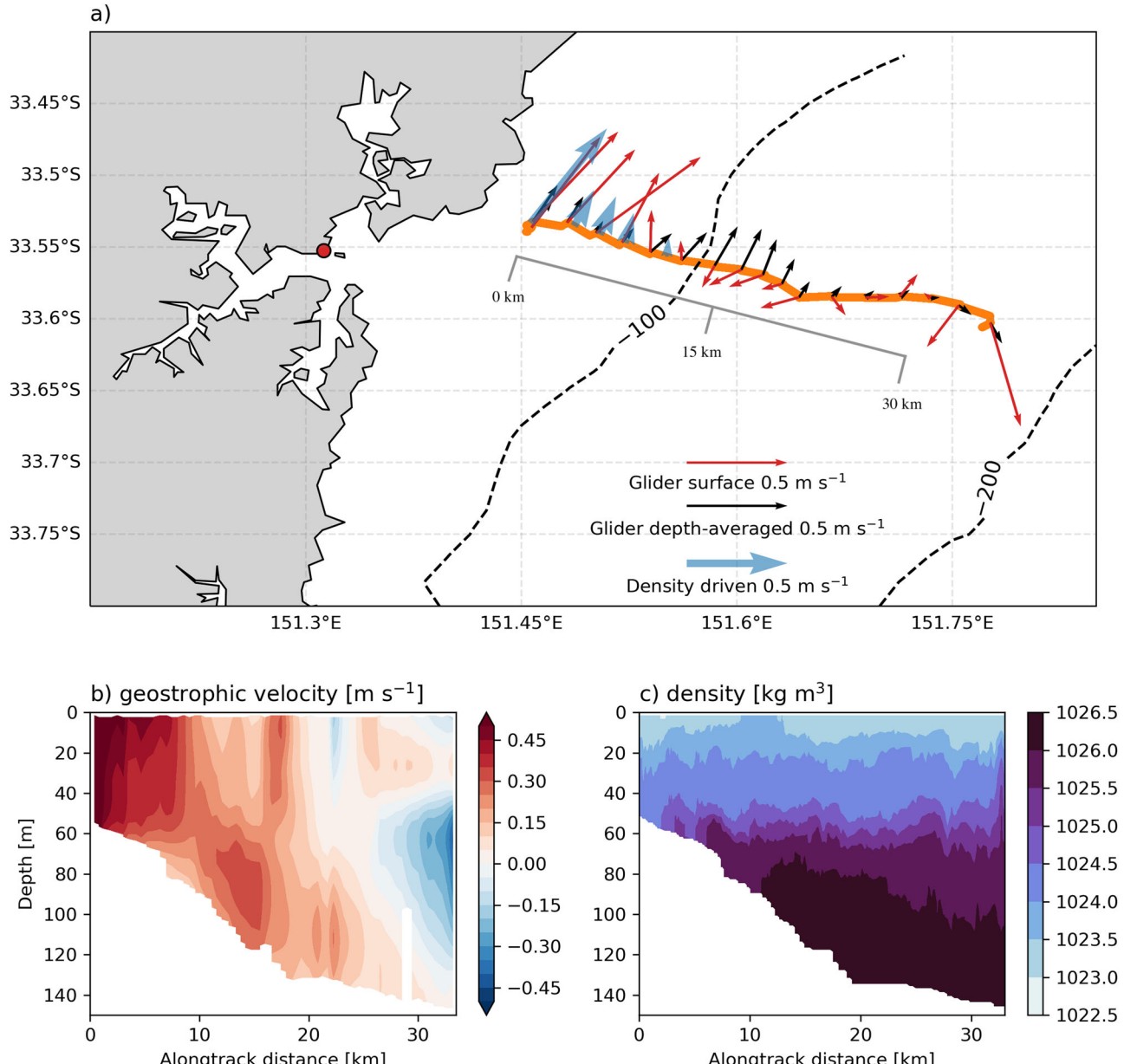

**Fig. 6 | The effect of estuarine discharge on coastal circulation. a** Depth-averaged (black vectors) and surface (red vectors) velocities observed by the glider during the offshore transect on 13 April 2022, compared to the buoyancy-driven component of the surface velocity (blue vectors) calculated from the observed density gradient. The red dot shows the location of the Hawkesbury River estuary data logger site. The grey line shows the distance scale for the panels below. **b** Alongshore geostrophic velocities, calculated perpendicular to the glider section from 13 April 2022, **c**) the density field sampled by the glider on 13 April 2022.

effect of fresher water being mixed downwards in the water column during the 'wet' missions is visible in increased spiciness (the variation of temperature and salinity along a constant density surface) in the top 80 m (Fig. 9d). The strong stratification set up by the homogeneous layer of freshwater observed in both of our previous case studies is clearly represented in the composite mean by the peak in $N^2$ at approximately 15 m. The difference between 'wet' and 'dry' glider missions is less clear when considering chlorophyll-a fluorescence, where 'wet' missions show lower chlorophyll and a shallower chlorophyll maximum when compared to 'dry missions'.

## Discussion

Sustained ocean observing with complementary platforms has allowed us to identify extreme salinity events in a complex continental shelf system. This is made possible by data from quarterly glider missions

that provide unprecedented spatial (horizontal and vertical) resolution through the water column, and a 13-year-long mooring timeseries of salinity, which allows us to categorise periods where salinity in shelf waters falls below the daily-varying 5th percentile as extreme low salinity events. Establishing this extreme threshold allows us to quantify the impact of the large freshwater flows into the coastal ocean during 2022, the wettest year on record in southeast Australia. In April 2022, extreme low salinity is widespread over the continental shelf between 32.5°S and 34°S. This extreme low salinity for the region is found as far as 70 km offshore, near the 1000 m isobath and persists almost unbroken for 2 months. This is more than five times greater than the offshore extent previously simulated in idealised modelling of an extreme event (12 km[12]).

We also observe a previously unseen dynamic for the region during April 2022: a 'double-stacked' stratification, with two layers of

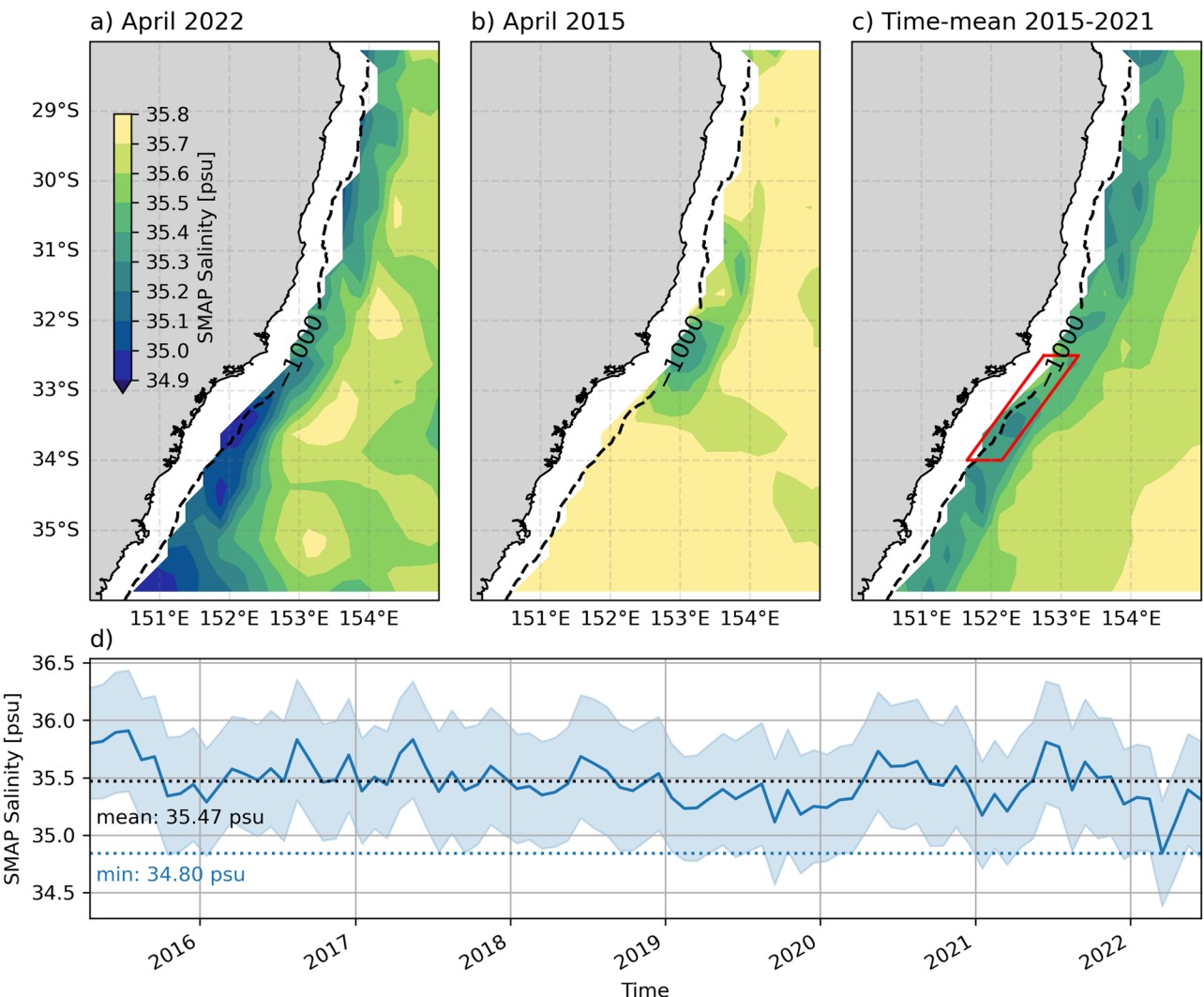

**Fig. 7 | Maps of monthly satellite salinity from the SMOS satellite product for. a)** April 2022, **b)** April 2015, and **c)** the time-mean for 2015-2021. Dashed lines show the position of the 1000 m isobath. The red box in panel c) shows the area averaged over for the timeseries shown in **d**), with shading showing the uncertainty in the satellite salinity measurement.

fresher water in the upper 50 m of the water column resulting in one salinity stratified layer, and another, deeper, temperature-stratified layer. Prior to the high rainfall accumulated during 2022, this 'double-stacking' had not been observed in this region, but appears to persist throughout 2022. We hypothesise that this is due to a newer layer of freshwater being added to the water column before the freshwater input from previous rainfall events has fully dissipated, resulting in persistent low salinity layers, as seen in the PH100 mooring time-series (Fig. 2b).

The 'double-stacking' of freshwater layers is not observed in the April 2015 case study, or in the composite mean of previous glider missions associated with heavy rainfall (Fig. 9). This could be as a result of the previous glider trajectories simply not having passed through areas of 'double-stacking', but due to the large amount of glider data that we have available and the spatial extent of the extreme low salinity conditions observed, we consider this unlikely. During 'wet' events prior to 2022 the freshwater layer is approximately 15 m thick, which is consistent with previous idea-lised modelling results of the region[12]. However, signs of 'double-stacked' stratification are visible in data during both the April case study and a subsequent glider mission in October 2022 (Supple-mentary information Fig. S1), despite freshwater plumes not being widespread enough for us to classify it as a 'wet' mission. If the

double-stacked stratification is indeed due to a build up of low salinity water from multiple large rainfall events, we would expect to see an increase in the residence times of extreme low salinity events in 2022. From the PH100 moored salinity timeseries we see that in 2022 there are extreme low salinity events up to 44 days long, while in the 12 years prior, the longest event was 8 days. Cumula-tively, there were 116 extreme low salinity days in 2022. This points to an increase in the residence time of extreme low salinity shelf water in 2022. Therefore, it appears that, within the 13 years for which data is available, the 'double-stacking' phenomena is unique to 2022. This begs the question whether we could see more 'double-stacked' freshwater layers in future if large rainfall events become more extreme, as is projected for the future[5].

The differing propagation directions of the plumes of low salinity water (Fig. 7) observed by satellite in April 2015 and April 2022 is also consistent with idealised modelling results[12], who found that the pre-sence of a cyclonic eddy at the shelf edge (as there is in our 2015 case study) results in northward plume propagation, whilst an extended EAC jet (as in our 2022 case study) results in southwestward plume propagation. Our results show that this mechanism, whereby the off-shore mesoscale flow controls the direction of plume propagation, is robust, even though the observed low salinity plumes are larger than those previously modelled.

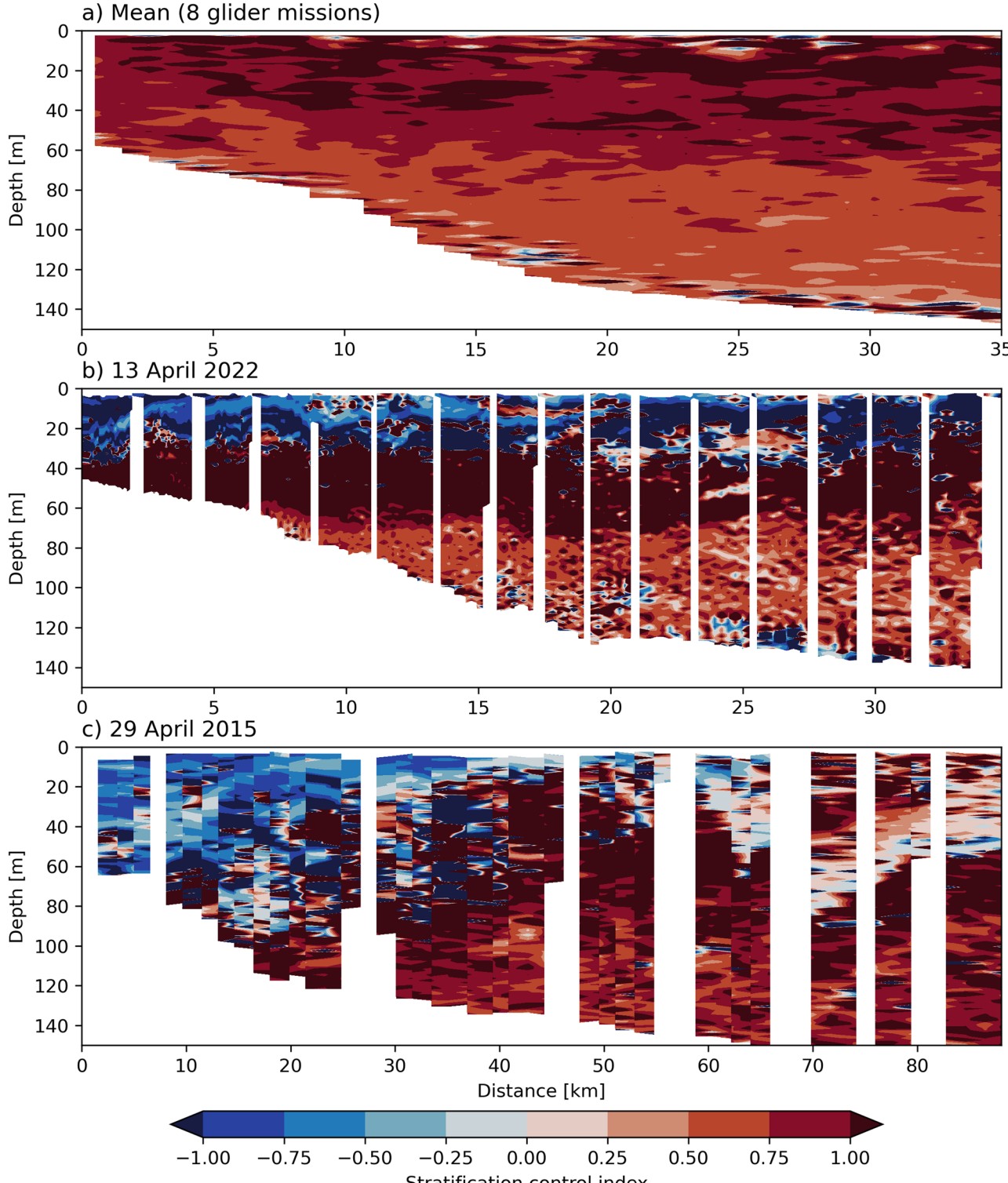

**Fig. 8 | Stratification control index (positive values show temperature stratified and negative values show salinity stratified waters) for. a)** a mean section at 33.5°S from 8 glider missions which performed transects during normal conditions, **b)** April 2022 low salinity event and **c)** April 2015 low salinity event.

Impacts of persistent extreme rainfall events on shelf ecosystems in this typically dry region are many. In addition to the direct impact on water quality from the dispersion of organic material from river mouths, there is evidence of mass kelp mortality caused by low salinity events[33]. Other impacts include a significant loss of optical depth, reducing light availability for primary production in an oligotrophic western boundary current system. Indeed, during the observed low salinity periods, the subsurface chlorophyll

maximum common to the region[34] appears to be suppressed. Low salinity events also impact habitat availability for ecosystems, with evidence of extreme rainfall affecting fisheries catches by shifting the species composition[35,36] and the distribution of larval fish[37]. Increases in the intensity of cross-shelf exchange, important for biological productivity in this region[29], have also been recorded in other western boundary currents affected by large freshwater discharges[9]. Thus, in a future climate, an increase in the frequency

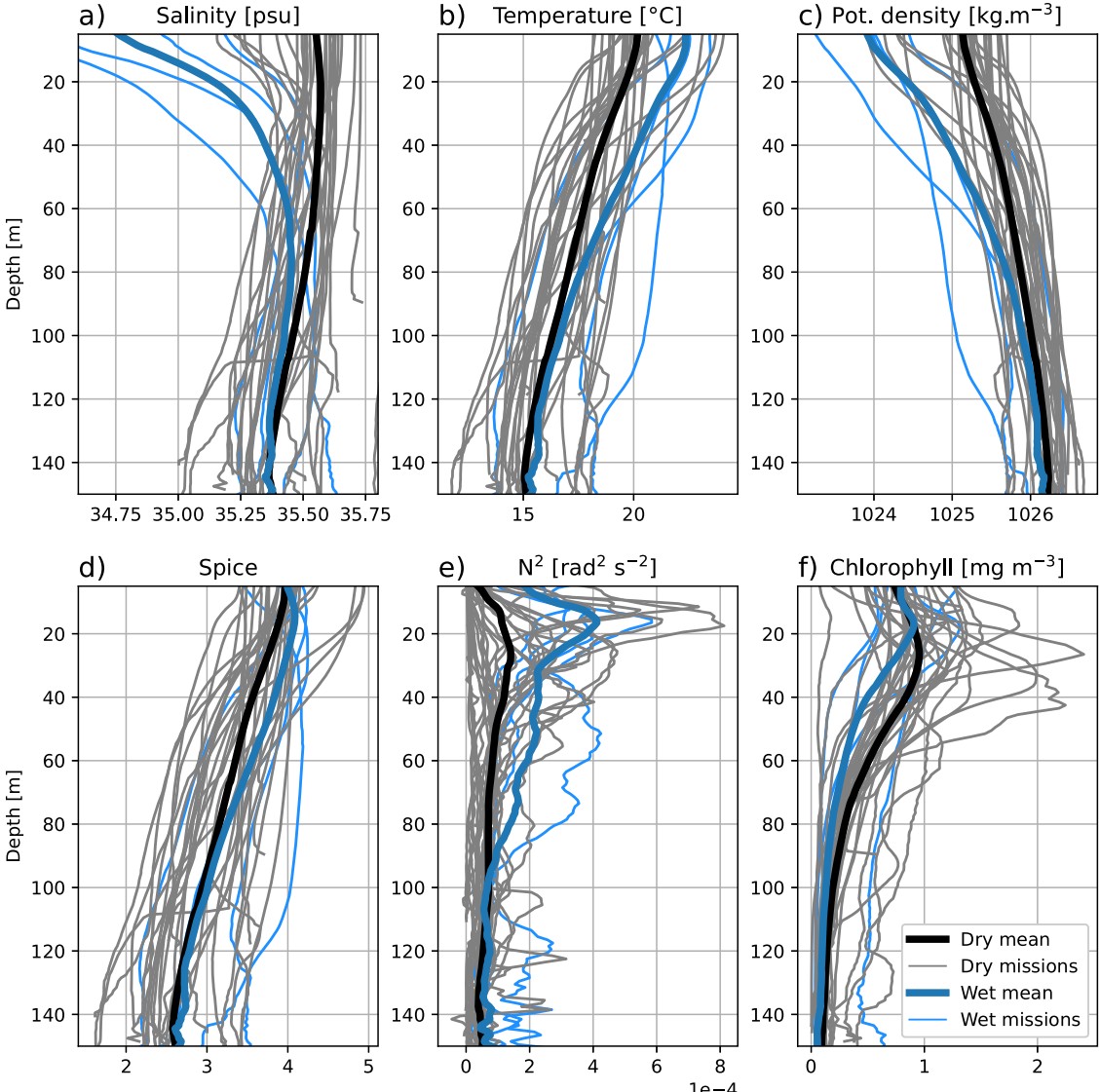

**Fig. 9 | Composite mean vertical profiles of. a)** salinity, **b)** temperature, **c)** potential density, **d)** spice, **e)** coloured dissolved organic matter (CDOM) and **f)** chlorophyll-a fluorescence, from glider missions between 32.5°S and 34°S and inshore of the 200 m isobath. Thick blue lines show the mean of extreme freshwater `wet' glider missions, while the thick black line shows the mean of all other missions. Thin blue and grey lines show the mean-profiles of individual `wet' and `dry' missions respectively.

of very large amounts of accumulated rainfall could disrupt fisheries catches and so impact food security. Before 2022, extreme low salinity events at the PH100 mooring tended to be short-lived with an average duration of 6 days (showing remarkable agreement with previous modelling results). However, in 2022 the duration increased, with the 4 longest extreme salinity events in the mooring record all occurring in 2022. Thus a period of unprecedented rainfall led to extreme low salinity conditions of long persistence, giving ecosystems little time to recover.

Increased stratification can lead to a reduced ocean-atmosphere heat flux and a shallower mixed layer depth, which has been shown to result in more surface-intensified marine heatwaves[38]. This appears to be the case off southeast Australia in 2022, where anecdotally the coast was affected by multiple marine heatwaves in the first half of the year. Thus, while at the basin scale, rainfall during La Niña years is driven by the transfer of heat and moisture from ocean to the atmosphere, at the shelf scale, a reduced ocean-atmosphere flux could lead to compound extremes in both salinity and temperature. Salinity extremes often co-occur with ocean temperature extremes[39] and these compound events deserve attention and investigation in the face of rapid environmental change.

More subtle impacts include the effect of freshwater discharge on surface frontogenesis and submesoscale variability on the shelf waters[40,41] by increasing both the lateral and vertical buoyancy gradients. Increased frontogenesis can impact the convergence of biological matter and pollutants. Both the 2022 and 2015 case studies feature sharp fronts between the shelf and offshore waters (see Figs. 4g and 5g), particularly the 2015 case where freshwater is subducted beneath the EAC forming a surface front, which exports potentially sediment-laden river water more than 100 km offshore. In the absence of any background flow, we would expect buoyant river plumes to flow equatorwards, as is the case for buoyant coastal currents such as the Norwegian Coastal Current[42]. In the Hawkesbury shelf case considered here, the equatorward flow tendency of the buoyant freshwater plume interacts with the background flow of the fast-flowing western boundary current. The interaction between the plume and EAC jet enhances the cross-shelf velocity gradient, therefore increasing frontogenesis between the shelf and offshore waters.

In a region where temperature is the dominant control on stratification, salinity has become an important control on stratification due to extreme rainfall. As rainfall extremes are predicted to increase, so could the importance and impacts of freshwater input on the productive shelf waters of this region. Currently, much of the world's coastal ocean does not have an ocean observing system capable of categorising and predicting extreme low salinity events. For example, in April 2022, concurrent with the floods in eastern Australia, extreme rainfall occurred in eastern South Africa[43], another region where it has previously been assumed that freshwater has little impact on continental shelf waters. However, without sustained subsurface salinity observations e.g. those provided by ocean gliders, the effects of extreme rainfall on the ocean cannot be quantified. These results show the advantage of a multi-platform approach to sustained ocean observing for identifying and characterising new dynamics and impacts of extreme events. The use of sustained glider and mooring observations in shelf regions globally would fill a crucial gap in our global ocean observing system[6] and allow the effects of extreme events in these crucial regions to be quantified as we adapt to life under a changing climate.

## Methods

### Rainfall data
Rainfall observations are taken from the Sydney Airport weather station (Bureau of Meteorology station ID:066037). This site is located at 33.95°S, 151.17°E at an elevation of 6 m above sea level. It is chosen because of its proximity to the Port Hacking 100 m oceanographic mooring site (PH100) used in this study to define extreme low salinity thresholds.

### Ocean gliders
Regular (nominally quarterly) ocean glider missions have been carried out along the inshore edge of the EAC (29.5-35°S) since 2008 as part of the Integrated Marine Observing System (IMOS)[22]. The gliders typically sample between the coast and the 200 m isobath from the surface to bottom, with 4-5 missions undertaken each year. Each mission lasts three to four weeks. This deployment strategy allows quasi-synoptic sampling of the shelf waters inshore of the EAC[22,44–46].

In this paper we use vertical hydrographic profiles taken during 36 glider missions over the shelf between 32.5°S and 34°S. Each mission provides between 3000-6000 quasi-synoptic individual hydrographic profiles from surface to near bottom, with the average distance between the glider surfacing being approximately 200 m. For more information on glider instrumentation and processing please see the supplementary information text.

### Satellite remote sensing
Visible colour images from Sentinel-2 were processed using tools from Digital Earth Australia[47]. Daily chlorophyll-a ocean colour estimates are obtained from the IMOS MODIS 1 day product at 4 km resolution, using the OC3 algorithm[48]. In coastal waters, chlorophyll concentrations derived from satellite ocean colour products are unreliable due to the complex interplay of chlorophyll, turbidity and coloured organic matter, hence we used ocean colour qualitatively to spatially examine front formation, rather than focusing on absolute values. Near-realtime surface geostrophic velocities (spatial resolution 0.2°) are obtained from an OceanCurrent product distributed by IMOS that merges satellite altimetry with sea level elevation measurements from coastal tide gauges[49] to improve accuracy in coastal regions.

### Port Hacking 100 mooring site
The Port Hacking mooring (PH100,[50]) is an oceanographic mooring site located in approximately 100 m of water at 34.12°S, 151.23°E (Fig. 1). It has measured salinity at a nominal depth of 15-24 m since 4th May 2010. The mooring provides a consistent, ~13-year long record of salinity at a fixed location, which makes it ideal for the determination of a salinity threshold for extreme freshwater events on the Hawkesbury shelf.

### Hawkesbury river estuary logger
A HOBO U26 CT data logger measuring near-surface temperature and salinity at a nominal depth of 1 m was placed at the mouth of the Hawkesbury River estuary (33.5523°S, 151.3124°S) from January to August 2022. Tidal variability is removed with a 48-hour lowpass filter.

### Creating a daily salinity climatology
Salinity measured at the Port Hacking 100 m site between 2009 and 2023 (~13 years) was used to create a daily climatology for identifying extreme low salinity events. We use the salinity mooring aggregated Long Time Series Product developed by the Australian National Mooring Network (ANMN) and the Australian Ocean Data Network (AODN)[51], that concatenates all deployment salinity files between 2009 and 2023 into one aggregated NetCDF file per site. Additionally, we use the IMOS CTD profiles collected since 2009[52].

We follow the method used by[50,53] to produce a daily salinity climatology at the Port Hacking 100 m site. Data were first quality controlled using standardised IMOS procedures[54–56], followed by additional quality control procedures as described by[50,53]. Considering data availability, and so to not bias the climatology statistics, we also exclude a small portion of data when salinity is extremely fresh (as identified in this study). Climatology statistics for each day were calculated at a depth of $16 \pm 3$ m (where most measurements are available) using data within a 11-day moving window, and as a final step were smoothed using a 31-day window, as suggested by[26].

### Estimating the duration of extreme freshwater events
We use the Port Hacking mooring salinity daily-varying 5th percentile at a depth of 16 m to identify days of extreme freshwater. We group consecutive days of extreme freshwater together as single events, and if two extreme freshwater events lasting $\geq 5$ days are separated by a less anomalous period of $\leq 2$ days, we combine these two freshwater events together, analogous to marine heatwave methodologies[26]. We then calculate duration as the total number of days in each extreme freshwater event.

### Estimation of density-driven currents
Density-driven currents are computed following[31] using a simple model allowing for the effects of rotation and friction[30]. This assumes that flow is forced by a depth uniform horizontal gradient in the across-shore direction and that there is no variation in the alongshore direction. Frictional stresses are controlled by eddy viscosity, giving equations of motion as

$$f\mathbf{v} - \frac{1}{\rho_0}\left(\frac{\partial p}{\partial x} + \frac{\partial \tau_x}{\partial z}\right) = f\mathbf{v} - g\frac{\partial \eta}{\partial x} + gz\frac{1}{\rho_0}\frac{\partial \rho}{\partial x} + N_z\frac{\partial^2 \mathbf{u}}{\partial z^2} = 0$$
$$-f\mathbf{u} - \frac{1}{\rho_0}\frac{\partial \tau_y}{\partial z} = -f\mathbf{u} + N_z\frac{\partial^2 \mathbf{u}}{\partial z^2} = 0. \tag{1}$$

We solve these numerically using routines supplied with[31]. The density gradient is taken from a straight line fitted to the vertically averaged upper 50 m density observed by the glider. The parameters used are shown in Table 1. Note that the density gradient used at the start of the transect is taken as the gradient between the Hawkesbury River estuary logger, and the beginning of the glider transect.

Full-depth absolute geostrophic currents for the April 2022 glider section were also computed and are shown in Fig. 6b. The dynamic height anomaly was first calculated using binned conservative temperature and absolute salinity measured by the glider and smoothed using a 40 dbar moving window over depth and a 0.2° moving window over longitude. Geostrophic velocities relative to the surface were then estimated from the dynamic height, and the absolute geostrophic

**Table 1 | Inputs used for estimating density-driven surface currents**

| Distance along transect [km] | 0 | 3 | 5 | 7 | 10 |
|---|---|---|---|---|---|
| h [m] | 20 | 50 | 70 | 80 | 100 |
| $\frac{\partial \rho}{\partial x}$ [kg m$^{-4}$] | 0.0015 | 9.3x10$^{-5}$ | 5x10$^{-5}$ | 3x10$^{-5}$ | 1.4x10$^{-5}$ |
| Eddy viscosity | 0.02 | 0.02 | 0.02 | 0.02 | 0.02 |

velocities were calculated by referencing the glider dive-averaged currents (depth-average current derived from the glider dead reckoning navigation and GPS fixes at the surface). The dynamic height and geostrophic velocity calculations were performed using the Gibbs Seawater Oceanographic toolbox[57].

## Temperature and salinity contributions to stratification

In order to quantify the contribution of extreme low salinity water to the stratification of the water column we use the stratification control index (SCI,[32]).

$$SCI = \frac{N_\theta^2 - N_S^2}{N_\theta^2 + N_\theta^2}, \tag{2}$$

$$N^2 = -\frac{g}{\rho_0}\frac{\partial \rho_\theta}{\partial z} = N_\theta^2 + N_S^2, \tag{3}$$

$$N_\theta^2 = g\alpha\frac{\partial \theta}{\partial z}, \tag{4}$$

$$N_S^2 = g\beta\frac{\partial S_A}{\partial z} \tag{5}$$

With $g$ being the gravitational acceleration as a function of latitude and pressure, $\alpha$ the thermal expansion coefficient of seawater and $\beta$ the saline contraction coefficient of seawater (all calculating following[57]). Where SCI $>= 1$, the ocean is temperature stratified, where SCI $<= -1$, the ocean is salinity stratified. If $-1 < SCI < 1$, the ocean is stratified by both temperature and salinity, known as transition. We expect the mid-latitude waters considered here in to be transitional waters, but the strength and sign of SCI provides an indication of whether the water column is dominantly stratified by temperature or salinity.

To quantify the SCI in the study region during normal conditions, we calculated a time-depth binned mean salinity transect using 8 glider missions. These 8 glider missions when combined consist of measurements mostly in September, November, and May in the years 2009, 2017, 2018, 2019, and 2020. These glider measurements were confined between 33.25 and 33.75 ° S within waters shallower than 200 m at times when salinity was not extreme. Time-depth bin averages based on 75 data points or more were used for the transect.

## Data availability

All data used in this study are freely available. Links for access can be found below. Rainfall: http://www.bom.gov.au/jsp/ncc/cdio/weatherData/av?p_nccObsCode=136&p_display_type=dailyDataFile&p_stn_num=066037&p_startYear=Visible colour satellite imagery: Contains modified Copernicus Sentinel data 2022 processed by Sentinel Hub https://scihub.copernicus.eu Altimetry[58]: https://thredds.aodn.org.au/thredds/catalog/IMOS/OceanCurrent/GSLA/DM/catalog.html Ocean colour[59]: http://thredds.aodn.org.au/thredds/catalog/IMOS/SRS/OC/gridded/aqua/catalog.html Glider[60]: http://thredds.aodn.org.au/thredds/catalog/IMOS/ANFOG/slocum_glider/catalog.html Data to recreate the figures in this paper can be found at[61] https://doi.org/10.5281/zenodo.10163560.

## Code availability

Code to perform the analyses and recreate the figures in this paper can be found at https://doi.org/10.5281/zenodo.10163560.

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

## Acknowledgements
The authors would like to acknowledge the work of the NSW-IMOS mooring team, the Australian National Facility for Ocean Gliders, Stuart Milburn, and all others involved in maintaining and funding these important timeseries. Data were sourced from Australia's Integrated Marine Observing System (IMOS) - IMOS is enabled by the National Collaborative Research Infrastructure Strategy (NCRIS). This work was partially funded by the NSW Government Marine Estate Management Strategy. Glider deployments in NSW were partially supported by the NSW Research Attraction and Acceleration Program.

## Author contributions
Conceptualisation: M.R., N.M. and T.I., Data analysis and processing: N.M. and M.H., N.M. wrote the initial draft. N.M., M.R., M.H. and T.I. developed and reviewed the final manuscript.

## Competing interests
The authors declare no competing interests.
