## [Peer Review File · Nature Communications]

Quantifying coastal freshwater extremes during unprecedented rainfall using long timeseries multi-platform salinity observationsREVIEWER COMMENTS

Reviewer #1 (Remarks to the Author):

The manuscript presents an unprecedented freshening event off eastern Australia with observations from underwater gliders. I think the manuscript presents relevant new information for salinity and density conditions under increased river runoff in the region. However, the study lacks crucial content about the impacts on a regional scale and the supposed new dynamics. It is very descriptive and a more in-depth analysis is required to fully evaluate their new findings.

The title fits with a relevant contribution in a Nature Communications paper. Nonetheless, the content of the manuscript is not really what the titles suggests. There is no "new dynamics" demonstrated and there are no major analyses of a decade of glider data. You really need to present further evidence in line with the title. For the coast of Australia a drop in salinity to about 34-34.5 psu might be very anomalous. However, for most coastal regions with freshwater influence those values would not imply a big change in buoyancy-driven currents and dynamics. What is the context of these "low salinity events" in the regional salinity budget of the region?. Can you quantify in relative terms the magnitude of the potential changes in coastal circulation and dynamics?, more than just showing salinity sections...

It would be helpful to evaluate and present river discharge data, more than just rainfall. What is the magnitude of the discharges during these anomalous events into the coastal ocean?. There could be lots of rain, but if that rain does not drain into the coastal ocean it is not very relevant for the coastal circulation.

2022 was an anomalous La Niña year according with rainfall. Ok, but how important is to consider an accumulated effect on the freshwater budget and dynamics in the coastal ocean?. There is mixing and the circulation should promote the transport of freshwater along the coast and offshore. Thus, the accumulated effect is relative to the residence time of that freshwater in the region. Can you comment on this?. If the freshwater and the circulation in the area is really important on the synoptic scale, then what is the importance of having an accumulated rainfall on the dynamics?. I think there is a serious issue of scales here.

What is the spatial variability of the freshwater conditions along the coast?. Could you present some validation of the cross-shore and alongshore salinity structure from the modeling studies in the region?. It is crucial to complement with modeling results as it is intended to explain a new dynamics. There are modeling efforts in which coauthors have participated. You should exploit the modeling results to really explore the anomalous conditions associated with these relative low-salinity events. There must be an important along-shelf variability in freshwater conditions and associated buoyancy-driven flows. The glider data would help to validate this freshwater events and to really report a new dynamics analyzing the modeling outputs. Even with the cross-shore glider sections the authors could compute buoyancy-driven flows, etc. You could also try to separate some terms of the forcing driving currents during these events. As is, the paper is too descriptive and lacks on any relevant dynamical analysis.

Considering the potential alongshore variability, the comparison of the cross-shore sections from 2015 and 2022 is not very consistent. How do you know that the difference in the layers captured on those events is not a function of the location along the coast?. Do you assume that the cross-shore structure evidenced on 2022 did not occurred in 2015 at all ?. The modeling results would offer strong evidence of this issue as well.

There are many typos throughout the manuscript. For example, the caption of figure 5 says "backscatter" for panel (d), but it is spice, no backscatter. It is necessary to have a careful revision before submitting a manuscript. Please be much more meticulous in your writing. There are some very weird sentences as "This is despite the exchange of freshwater to and from the ocean from land...". Please, the writing must be way better.

Based on the information from section 4.3. Why not using fluorescence images then?. If chlorophyll images are not supposed to be of good quality then a comparison with fluorescence should be shown.

Section 4.5. seems to be wrong in the compilation. The references are not properly cited.

Figure A1 does not have the letters according with the caption. How is this possible?.

The list of references must be checked and corrected accordingly. Jones et al. is a Nature paper?. NLM Medline does not seem to be correct. In reference number 22 there are some question marks ????. Some journals are included in their full extension, others abbreviated. Please be consistent. Reference number 31 does not have the journal name, and presents 2 doi numbers, etc.

I hope the authors take these comments in consideration seriously. The manuscript has the potential to report relevant results (not just in the local context) but all these comments and issues must be fully considered and resolved.

The best.

Reviewer #2 (Remarks to the Author):

The study examines extreme rainfall events across southeast Australia and explores the concept of using ocean salinity as an indicator of the ocean extreme events. The utility of salinity-derived indicators of the extreme events adds value to the established framework of temperature-defined ocean extremes, expanding its applicability to high precipitation/repeated flooding events. The paper is well written, the main conclusions are well articulated, and it would be of interest to a wide audience of Earth and ocean scientists. I do have a couple of concerns that I recommend the authors to think about and address in their discussion and interpretation:

1. When you introduce low-salinity events as a direct fingerprint of rain events, what is your evidence of the direct link between the two? How did you identify that ALL low-salinity signatures are indeed rain-induced fresh lenses, sitting in the ocean long enough to be captured by the instruments? How do you separate the impact of rain (forcing) and strong advection or mixing (fluxes)? Do rain events last a long time that mixing and advection never catch up? It would be more convincing to see dynamical analysis of low-salinity signatures and other factors, to clearly link freshening to rain events, which is the main goal of this framework?

2. I find it a remiss not to use direct salinity observations from the satellites, and use salinity proxy (like CDOM) instead. I recommend exploring salinity signatures in actual salinity datasets, like NASA's SMAP or ESA's SMOS missions, both overlap with the study period. It would add value and robustness to your conclusions to use salinity as an indicator of extreme events, like precipitation and flooding.

3. I found following Figure 2 somewhat difficult and not intuitive. Is it possible to add a panel illustrating a potential relationship between rain events and freshening events? I couldn't quite follow the linkage between the accumulated rain and salinity changes in panel (b).

4. The depth of rain-influenced freshening was interesting! Again, it would be great to add some dynamical context and see basic budget analysis to see the interplay of the underlying physical processes to know for sure what drives the deepening of the surface fresh lenses into the deep; what processes support 'double-stacking' hypothesis and how persistent is it?

Response to reviewers for: "Unprecedented rainfall drives coastal freshwater extremes and new dynamics: Insight from a decade of autonomous ocean glider observations"

Reviewer #1 (Remarks to the Author):

The manuscript presents an unprecedented freshening event off eastern Australia with observations from underwater gliders. I think the manuscript presents relevant new information for salinity and density conditions under increased river runoff in the region. However, the study lacks crucial content about the impacts on a regional scale and the supposed new dynamics. It is very descriptive and a more in-depth analysis is required to fully evaluate their new findings.

Thank you, we are glad that you find this study relevant, on your and another reviewer's recommendation, we have added new analyses using both satellite salinity and more in-depth examination of the glider data. This has allowed us to examine both regional scale impacts as well as quantifying the effect of the freshwater discharge on both the shelf circulation and vertical stratification. Please see below for specifics.

The title fits with a relevant contribution in a Nature Communications paper. Nonetheless, the content of the manuscript is not really what the titles suggests. There is no "new dynamics" demonstrated and there are no major analyses of a decade of glider data. You really need to present further evidence in line with the title. For the coast of Australia a drop in salinity to about 34-34.5 psu might be very anomalous. However, for most coastal regions with freshwater influence those values would not imply a big change in buoyancy-driven currents and dynamics. What is the context of these "low salinity events" in the regional salinity budget of the region?. Can you quantify in relative terms the magnitude of the potential changes in coastal circulation and dynamics?, more than just showing salinity sections...

We agree that while the drop in salinity that we observe here is extreme for this region, a salinity difference of similar magnitude may not be anomalous in other regions with higher freshwater influence. This was one of the reasons why we adopted a percentile framework, and chose a percentile threshold based on the observed salinity distribution in our study area. This was perhaps not properly emphasised in the manuscript, and so we have now added to the following text to section 2.1:

"The use of a seasonally varying percentile threshold, rather than a fixed one, as well as the use of the observed salinity distribution to guide the choice of that percentile threshold, is important in choosing appropriate extreme salinity values for the region. In a region of low salinity variability, a relatively small change in salinity could be considered extreme, while in a region of high salinity variability a larger drop in salinity would be needed to be categorised as extreme."

We have removed reference to new dynamics in the title, which now reads "Quantifying coastal freshwater extremes during unprecedented rainfall: Insight from over a decade of multiplatform salinity observations"

As suggested, we have quantified the impact of the freshwater inflow on the coastal circulation by making greater use of the glider observations. Following Heaps (1972) and Simpson and Sharples (2012), we calculate the buoyancy-driven currents for the April 2022 case study and compare them to the observed surface and depth-averaged currents from the glider. This analysis shows northeastward velocities > 0.5 m/s driven by the freshwater input from the Hawkesbury River. These results are presented in section 2.4 of the revised manuscript and in the new Figure 6 (shown below).

Figure 6: Depth-averaged (black vectors) and surface (red vectors) velocities observed by the glider during the offshore transect on 13 April 2022, compared to the buoyancy driven component of the surface velocity (blue vectors) calculated from the observed density gradient. The red dot shows the location of the Hawkesbury River estuary data logger site. It would be helpful to evaluate and present river discharge data, more than just rainfall. What is the magnitude of the discharges during these anomalous events into the coastal ocean?. There could be lots of rain, but if that rain does not drain into the coastal ocean it is not very relevant for the coastal circulation.

River discharge for this region is not well known. Hence, the previous modelling study relied on estimated, idealised discharge volumes. This is an issue for future modelling studies, and we hope that this paper will motivate an improvement of the hydrological modelling of this region. For this paper we have been able to draw upon a logger mounted in the mouth of the Hawkesbury, the largest river in the region. We have included a new figure 3 and added section 2.2, which shows the relationship between rainfall, estuarine inflow salinity, and salinity from the PH100 coastal mooring. Estuarine salinity (which drops to 0 during large rainfall events) and coastal salinity from the mooring (65 km away) show a strong correlation ($r = 0.73$ at 1 day lag), with minima occurring following the 3 large rainfall events of 2022.

Figure 3: Timeseries from January to August 2022 of a) surface salinity in the mouth of the Hawkesbury River estuary, b) near-surface salinity at the PH100 mooring site and c) rainfall at Sydney Airport. Grey shading shows the time extent of glider missions in the Hawkesbury shelf region during 2022.

2022 was an anomalous La Niña year according with rainfall. Ok, but how important is to consider an accumulated effect on the freshwater budget and dynamics in the coastal ocean?. There is mixing and the circulation should promote the transport of freshwater along the coast and offshore. Thus, the accumulated effect is relative to the residence time of that freshwater in the region. Can you comment on this?. If the freshwater and the circulation in the area is really important on the synoptic scale, then what is the importance of having an accumulated rainfall on the dynamics?. I think there is a serious issue of scales here.

We have addressed this issue of whether the accumulated freshwater discharge is large enough to actually be important in 3 ways:

- Firstly, we have used the stratification control index (Caneill et al. 2022, see appendix C) to show how the stratification of shelf waters becomes salinity-controlled during extreme rainfall events in our 2022 and, to a lesser extent, 2015 case studies. This analysis is discussed in the new section 2.6, as well as the new figure 8. This new analysis provides support for the statement made in the discussion, that: “In a region where temperature is the dominant control on stratification salinity has become an important control on stratification due to extreme rainfall. As rainfall extremes are predicted to increase, so could the importance and impacts of freshwater input on the shelf waters of this region.”

Figure 8: Stratification control index (positive values show temperature stratified and negative values show salinity stratified waters) for a) a mean section at 33.5°S from 8 glider missions which performed transects during normal conditions, b) April 2022 low salinity event and c) April 2015 low salinity event.

- Secondly, we use the observed glider section from April 2022 to show the density driven velocities due to the freshwater plume, that these are a large part of the inner and mid-shelf circulation, and that they form a velocity front with the dominant poleward flow offshore (new figure 6 and section 2.4)
- Thirdly, to address the residence time and accumulated impact of freshwater discharge from repeated extreme rainfall events, we have added a section to the

discussion on the increase in the number of low salinity days and length of events under the extreme rainfall conditions of 2022. It reads: "If the double-stacked stratification is indeed due to a build up of low salinity water from multiple large rainfall events, we would expect to see an increase in the residence of extreme low salinity events in 2022. From the PH100 moored salinity timeseries we see that in 2022 there are extreme low salinity events up to 44 days long, while in the 12 years prior, the longest event was 8 days. Cumulatively, there are 116 extreme low salinity days in 2022. This points to an increase in the residence time of extreme low salinity shelf water in 2022."

What is the spatial variability of the freshwater conditions along the coast?. Could you present some validation of the cross-shore and alongshore salinity structure from the modeling studies in the region?. It is crucial to complement with modeling results as it is intended to explain a new dynamics. There are modeling efforts in which coauthors have participated. You should exploit the modeling results to really explore the anomalous conditions associated with these relative low-salinity events. There must be an important along-shelf variability in freshwater conditions and associated buoyancy-driven flows. The glider data would help to validate this freshwater events and to really report a new dynamics analyzing the modeling outputs. Even with the cross-shore glider sections the authors could compute buoyancy-driven flows, etc. You could also try to separate some terms of the forcing driving currents during these events. As is, the paper is too descriptive and lacks on any relevant dynamical analysis.

At the suggestion of reviewer 2, we have included analysis of satellite salinity for our 2 case studies, which allows us to confirm that the EAC's control on mesoscale propagation (reported by Li et al. 2022) is consistent with our 2015 and 2022 case studies.

As we do not have realistic modelling outputs available due to the lack of freshwater discharge data, we believe it is best not to perform additional model analysis. A modelling analysis would be interesting, but due to the estimations involved in attempting to get the right discharge, it would involve many sensitivity studies – which we believe is beyond the scope of the paper. We have included additional comparison of our results with the previous modelling study (Page 14, paragraph 3 of the revised manuscript) and as suggested have computed buoyancy driven flows from the glider data.

This paper is by its nature a descriptive one detailing a recent extreme event, and thus we have removed reference to 'new dynamics' from the title. Detailed work on the dynamics, salinity budgets, vertical mixing etc. would be better placed in a second, more technical paper.

Considering the potential alongshore variability, the comparison of the cross-shore sections from 2015 and 2022 is not very consistent. How do you know that the difference in the layers captured on those events is not a function of the location along the coast?. Do you assume that the cross-shore structure evidenced on 2022 did not occurred in 2015 at all ?. The modeling results would offer strong evidence of this issue as well.

Please see context provided by satellite data in the new Figure 7 (shown below in response to reviewer 2)and section 2.5.

There are many typos throughout the manuscript. For example, the caption of figure 5 says "backscatter" for panel (d), but it is spice, no backscatter. It is necessary to have a careful revision before submitting a manuscript. Please be much more meticulous in your writing. There are some very weird sentences as "This is despite the exchange of freshwater to and from the ocean from land...". Please, the writing must be way better.

The manuscript has been revised and typos corrected. The sentence mentioned now reads “This is despite 80% of global surface freshwater fluxes taking place over the ocean...”

Based on the information from section 4.3. Why not using fluorescence images then?. If chlorophyll images are not supposed to be of good quality then a comparison with fluorescence should be shown.

As stated in the paper methods, we are using satellite chlorophyll as a tracer, hence the difficulty in distinguishing between turbidity and chlorophyll is not problematic. We have rephrased that sentence in section 4.3 which now reads: “In coastal waters, chlorophyll concentrations derived from satellite ocean colour products is unreliable due to the complex interplay of chlorophyll, turbidity and coloured organic matter, hence we used ocean colour qualitatively to spatially examine front formation, rather than focusing on absolute values.”

Section 4.5. seems to be wrong in the compilation. The references are not properly cited. Reference citations have been corrected.

Figure A1 does not have the letters according with the caption. How is this possible?. This oversight has been corrected.

The list of references must be checked and corrected accordingly. Jones et al. is a Nature paper?. NLM Medline does not seem to be correct. In reference number 22 there are some question marks ????. Some journals are included in their full extension, others abbreviated. Please be consistent. Reference number 31 does not have the journal name, and presents 2 doi numbers, etc. References have been checked and corrected.

I hope the authors take these comments in consideration seriously. The manuscript has the potential to report relevant results (not just in the local context) but all these comments and issues must be fully considered and resolved. The manuscript has become significantly longer and more in-depth, especially in the areas which you have highlighted.

The best.

Reviewer #2 (Remarks to the Author):

The study examines extreme rainfall events across southeast Australia and explores the concept of using ocean salinity as an indicator of the ocean extreme events. The utility of salinity-derived indicators of the extreme events adds value to the established framework of temperature-defined ocean extremes, expanding its applicability to high precipitation/repeated flooding events. The paper is well written, the main conclusions are well articulated, and it would be of interest to a wide audience of Earth and ocean scientists. I do have a couple of concerns that I recommend the authors to think about and address in their discussion and interpretation:

Thank you for your recognition of the utility of our framework for salinity extremes. We have addressed your concerns below by including additional datasets and analysis in the revised manuscript.

1. When you introduce low-salinity events as a direct fingerprint of rain events, what is your evidence of the direct link between the two? How did you identify that ALL low-salinity signatures

are indeed rain-induced fresh lenses, sitting in the ocean long enough to be captured by the instruments? How do you separate the impact of rain (forcing) and strong advection or mixing (fluxes)? Do rain events last a long time that mixing and advection never catch up? It would be more convincing to see dynamical analysis of low-salinity signatures and other factors, to clearly link freshening to rain events, which is the main goal of this framework?

To strengthen this link we have introduced a new figure 3, which shows the relationship between salinity at the mouth of the Hawkesbury (the largest estuary in the region), coastal salinity from a mooring (65 km south from the estuary) and rainfall from nearby Sydney airport. For when data is available, which is January to August 2022, the correlation between salinity at the Hawkesbury estuary mouth (a proxy for freshwater input) and salinity at the mooring site is 0.73 at a 1 day lag. Salinity minima at both the estuary mouth and the coastal mooring follow on from large rainfall events in late February/early March, early April and early July. The new analysis is discussed in the new section 2.2.

Figure 3: Timeseries from January to August 2022 of a) surface salinity in the mouth of the Hawkesbury River estuary, b) near-surface salinity at the PH100 mooring site and c) rainfall at Sydney Airport. Grey shading shows the time extent of glider missions in the Hawkesbury shelf region during 2022.

2. I find it a remiss not to use direct salinity observations from the satellites, and use salinity proxy (like CDOM) instead. I recommend exploring salinity signatures in actual salinity datasets, like NASA's SMAP or ESA's SMOS missions, both overlap with the study period. It would add value and robustness to your conclusions to use salinity as an indicator or extreme events, like precipitation and flooding.

Thank you for this suggestion, we had not considered the use of satellite salinity due to the coastal nature of the study. However, upon exploring the SMAP product you suggest, we find that the low salinity signatures extend further offshore than previously thought. We have included a new figure (figure 7, copied below) showing the large-scale salinity signatures associated with our April 2022 and 2015 case studies, as well as a time-mean between 2015 and 2021, and a timeseries which shows the general salinity minima which we associate with the extreme rainfall in April 2022. The propagation of the low salinity signature (southward in 2022, northward in 2015) is also consistent with previously idealised modelling results.

A new section 2.5 titled "Spatial extent of low salinity impact" has been added to the result section. And the figure shown below is the new Figure 7.

Figure 7: Maps of monthly satellite salinity from SMOS for a) April 2022, b) April 2015, and c) the time-mean for 2015-2021. Dashed lines show the position of the 1000m isobath. The red box in panel c) shows the area averaged over for the timeseries shown in d), with shading showing the uncertainty in the satellite salinity measurement.

3. I found following Figure 2 somewhat difficult and not intuitive. Is it possible to add a panel illustrating a potential relationship between rain events and freshening events? I couldn't quite follow the linkage between the accumulated rain and salinity changes in panel (b).

We have added a new figure (Fig. 3, shown in response above) that directly shows the co-variability between rainfall, estuarine inflow salinity, and coastal salinity. We believe that this makes the relationship between rainfall and low salinity events clear. We have also simplified and added some new annotations to Fig. 2 panel A to improve clarity.

4. The depth of rain-influenced freshening was interesting! Again, it would be great to add some dynamical context and see basic budget analysis to see the interplay of the underlying physical processes to know for sure what drives the deepening of the surface fresh lenses into the deep; what processes support 'double-stacking' hypothesis and how persistent is it?

We are glad that you find it interesting – the deeper fresh layer which we observed was unexpected, and no doubt rather complex. A budget analysis is hindered by a couple of factors, firstly that the freshwater discharge in the region is unknown, and secondly the very high energy circulation associated with the EAC and its mesoscale eddy field. The alternative approach we have taken is to use the stratification control index (Caneill et al. 2022, see appendix C) applied to the glider data. The analysis shows how the stratification of the shelf waters becomes salinity-controlled during

extreme rainfall events in 2022 and, to a lesser extent, the 2015 conditions compared to a mean state which is completely temperature controlled. The results are discussed in the new section 2.6, as well as the new figure 8. This new analysis provides support for the statement made in the discussion, that: “In a region where temperature is the dominant control on stratification, salinity has become an important control on stratification due to extreme rainfall. As rainfall extremes are predicted to increase, so could the importance and impacts of freshwater input on the shelf waters of this region.”

Figure 8: Stratification control index (positive values show temperature stratified and negative values show salinity stratified waters) for a) a mean section at 33.5°S from 8 glider missions which performed transects during normal conditions, b) April 2022 low salinity event and c) April 2015 low salinity event.

REVIEWER COMMENTS

Reviewer #1 (Remarks to the Author):

The new revised version of the manuscript is considerably better than the original draft. I do believe the authors have improved it significantly. The new sections and figures help to further understand the anomalous freshwater conditions in the study area. Conditions which could become more frequent under a changing climate, and high rainfall events along eastern Australia.

The new figures lead me to some new doubts though. These are minors issues that the authors could resolve/clarify relatively quickly. I suggest to change the status of the paper to "minor revisions" until this points are clarified.

1. The new Figure 6 provides some new insights on the dynamics associated with buoyancy-driven flows. I miss a cross-shore sections with the vertical structure of the buoyancy-driven (geostrophic velocities) alongshore currents and density field. Please show the cross-shore sections as a second and third panels of Figure 6. This will help to evidence the vertical structure of the alongshore flow and its offshore variability.

2. I agree with reviewer 2 on the use of satellite-derived salinity observations to provide some spatial context of the freshening events. The new Figure 7 gives good insights on this. However, it would be much better to compute the EOFs to really assess some spatial-temporal pattern associated with these anomalous freshening. The temporal variability of the dominant modes (PCs) could also be included to validate anomalous salinity signatures during the dates of anomalous rainfall. If this new figure provides better information, then, this average panels could be included as supplementary information. An EOF is a more robust analysis to show spatial-temporal variability.

I agree that diving into the dynamics could be out of the scope of this paper. However, it would have made it much more cited. I hope the authors could continue with those analysis in a future contribution.

The best.

Reviewer #2 (Remarks to the Author):

I would like to thank the authors for carefully addressing my questions and suggestions, adding relevant figures and explanation. Addition of satellite salinity plots in new Figure 7 was particularly illuminating. I recommend this manuscript be accepted for publications.

Reviewer #1 (Remarks to the Author):

The new revised version of the manuscript is considerably better than the original draft. I do believe the authors have improved it significantly. The new sections and figures help to further understand the anomalous freshwater conditions in the study area. Conditions which could become more frequent under a changing climate, and high rainfall events along eastern Australia.

The new figures lead me to some new doubts though. These are minors issues that the authors could resolve/clarify relatively quickly. I suggest to change the status of the paper to "minor revisions" until this points are clarified.

1. The new Figure 6 provides some new insights on the dynamics associated with buoyancy-driven flows. I miss a cross-shore sections with the vertical structure of the buoyancy-driven (geostrophic velocities) alongshore currents and density field. Please show the cross-shore sections as a second and third panels of Figure 6. This will help to evidence the vertical structure of the alongshore flow and its offshore variability.

Thank you for your considered input in improving the manuscript. As suggested, we have added geostrophic velocities and the density field from the glider section as second and third panels below figure 6. We have shown the new figure below:

Fig. 6 a) Depth-averaged (black vectors) and surface (red vectors) velocities observed by the glider during the offshore transect on 13 April 2022, compared to the buoyancy driven component of the surface velocity (blue vectors) calculated from the observed density gradient. The red dot shows the location of the Hawkesbury River estuary data logger site. The grey line shows the distance scale for the panels below. b) Alongshore geostrophic velocities, calculated perpendicular to the glider section from 13 April 2022, c) the density field sampled by the glider on 13 April 2022.

The following text has been added to section 2.4:

“Full-depth geostrophic velocities (Fig. 6b), are also calculated from the glider density field (Fig. 6c) to show the vertical structure. Although the utility of geostrophy can be limited this close to the coast, we see that the northeastward inshore flow associated with the freshwater plume, while surface-intensified, extends to a near-bottom depth of 50 m. The northeastward plume flow shallows and weakens in the offshore direction. The effect of the poleward flowing EAC can be seen at the very offshore end of the transect (Fig. 6b).”

We have explained the methods used by adding the following text to Appendix A:

“Full-depth absolute geostrophic currents for the April 2022 glider section were also computed and are shown in Fig. 6b. The dynamic height anomaly was first calculated using binned conservative temperature and absolute salinity measured by the glider and smoothed using a 40 dbar moving window over depth and a 0.2° moving window over longitude. Geostrophic velocities relative to the

surface were then estimated from the dynamic height, and the absolute geostrophic velocities were calculated by referencing the glider dive-averaged currents (depth-average current derived from the glider dead reckoning navigation and GPS fixes at the surface). The dynamic height and geostrophic velocity calculations were performed using the Gibbs Seawater Oceanographic toolbox (McDougall et al. 2011).”

2. I agree with reviewer 2 on the use of satellite-derived salinity observations to provide some spatial context of the freshening events. The new Figure 7 gives good insights on this. However, it would be much better to compute the EOFs to really assess some spatial-temporal pattern associated with these anomalous freshening. The temporal variability of the dominant modes (PCs) could also be included to validate anomalous salinity signatures during the dates of anomalous rainfall. If this new figure provides better information, then, this average panels could be included as supplementary information. An EOF is a more robust analysis to show spatial-temporal variability. We are glad that Figure 7 provides good insights. To explore further, as suggested we have computed EOFs from the satellite salinity data. However, these have not proved useful for validating the anomalous salinity signature, as the first EOF only explains 6% of the variance. We believe this is because EOF's are best used for identifying oscillating patterns, whereas in this region the salinity is affected by extreme rainfall events, which do not occur on regular intervals. Added to this, we have to keep in mind that this dataset is limited in length (7 years) and has relatively high uncertainty (0.5psu). Thus, while it is valuable for providing spatial context as in the current Figure 7, the addition of EOFs does not provide better information, and so we have not included them as supplementary information.

I agree that diving into the dynamics could be out of the scope of this paper. However, it would have made it much more cited. I hope the authors could continue with those analysis in a future contribution.

We certainly hope to in the future.

The best.

Reviewer #2 (Remarks to the Author):

I would like to thank the authors for carefully addressing my questions and suggestions, adding relevant figures and explanation. Addition of satellite salinity plots in new Figure 7 was particularly illuminating. I recommend this manuscript be accepted for publications.

Thank you for your valuable input in improving the manuscript.

REVIEWERS' COMMENTS

Reviewer #1 (Remarks to the Author):

I am glad the authors have addressed my comments. I believe the new version of the paper deserves to be published. The addition of the glider-derived geostrophic velocities is key for the comparison of the velocity field and the cross-shore structure of the circulation during that event.

I look forward for the final published version. Well done.

The best.